# Role of Exosomal miRNA in Bladder Cancer: A Promising Liquid Biopsy Biomarker

**DOI:** 10.3390/ijms22041713

**Published:** 2021-02-08

**Authors:** Xuan-Mei Piao, Eun-Jong Cha, Seok Joong Yun, Wun-Jae Kim

**Affiliations:** 1Department of Urology, College of Medicine, Chungbuk National University, Cheongju 28644, Korea; phm1013@hotmail.com (X.-M.P.); sjyun@chungbuk.ac.kr (S.J.Y.); 2Department of Biomedical Engineering, College of Medicine, Chungbuk National University, Cheongju 28644, Korea; ejcha@chungbuk.ac.kr; 3Institute of Urotech, Cheongju 28120, Korea; 4Department of Urology, Chungbuk National University Hospital, Cheongju 28644, Korea

**Keywords:** bladder cancer, liquid biopsy, exosome, exosomal miRNAs

## Abstract

Bladder cancer (BCa) is the most prevalent neoplasia of the urinary tract. Unfortunately, limited improvements in effective BCa management have meant that it remains a challenging disease. Cystoscopy has been the gold standard for BCa diagnosis and surveillance for over two centuries but is an invasive and expensive approach. Recently, liquid biopsy has been identified as a promising field of cancer research, due to its noninvasiveness and ease of sampling. Liquid biopsy samples could provide comprehensive information regarding the genetic landscape of cancer and could track genomic evolution of the disease over time. Exosomes, which contain RNAs, DNAs, and proteins, are a potential source of tumor biomarkers in liquid biopsy samples. In particular, exosomal miRNAs (exomiRs) hold great promise as biomarkers for tumor development and progression. In this review, we provide an overview of liquid biopsy biomarkers, with a particular focus on the use of exomiRs as biomarkers of cancer, and summarize their clinical implications for BCa. Finally, we discuss the future perspectives of these biomarkers in cancer research.

## 1. Liquid Biopsy in Bladder Cancer

### 1.1. Current Landscape

An estimated 1,806,590 new cases of cancer will be diagnosed in the United States in 2020, and an estimated 606,520 people will die from the disease [1]. Although progresses in immuno-oncology and personalized medicine have the potential to reduce the number of cancer deaths, the lack of early diagnostic tests to detect the presence of cancer before the onset of physiological symptoms remains an urgent issue. For decades, a combination of cytology and cystoscopy has been used routinely for the diagnosis, prognosis, and surveillance of bladder cancer (BCa); however, several outstanding issues remain. Specifically, the poor sensitivity of cytology in detecting BCa, the high invasiveness of cystoscopy, and the substantial inter- and intra-observer variations in tumor stage and grade interpretation emphasize the urgent need for improvements in BCa clinical guidance [2]. 

Over the last twenty years, the use of liquid biopsies as a noninvasive way to characterize the genomic landscape of cancer patients has been increasing steadily. The term “liquid biopsy” includes the sampling and analysis of biological fluids such as blood, plasma, urine, pleural liquid, cerebrospinal fluid, and saliva (Figure 1) [3,4]. The main focus of this research has been on circulating tumor cells (CTC) and circulating cell-free tumor DNA (ctDNA), but additional biomarkers present in the blood or urine, such as exosomes, could also be used for early cancer detection. The presence of cell-free DNA and RNA in human blood was first demonstrated in 1948 [5]; however, only a few liquid biopsies are currently approved for clinical use. The identification of suitable biomarkers for the presence and prognosis of a disease, as well as for monitoring the effect of certain treatments, is still a hot topic in cancer research. In this review, we discuss the benefits of liquid biopsy in comparison to tissue biopsy and summarize the roles of exosomal miRNAs (exomiRs) in the diagnosis and monitoring of BCa.

### 1.2. Properties of Liquid Biopsy

At present, the standard evaluation of cancer diagnosis or prognosis relies on a solid tissue biopsy and imaging. However, these techniques are limited by the size of the tumor, the invasiveness of the procedure, and operator-dependent variability in the sample collection and analysis. In particular, the heterogeneity of solid tumors can lead to variable results, depending on the specific site of tissue biopsy [6,7]. Tumor heterogeneity is a hallmark of BCa that occurs on multiple levels and directly affects clinical care [8]. As an alternative approach to overcome the problems associated with solid tissue biopsy, liquid biopsy is a minimally invasive procedure that allows the collection of samples at any time and can be used for serial monitoring. In particular, liquid biopsy can be used to generate a complete and real-time molecular profile of the genomic alterations arising from multiple regions of primary and metastatic tumors. Thus, this technique could overcome the limitations of tissue biopsy, which provides only a snapshot of tumor heterogeneity, depending on when and where the biopsy was obtained. As urine is stored in the bladder and would therefore be expected to contact a bladder tumor directly, it is a bona fide liquid biopsy sample for the detection of BCa. Overall, liquid biopsy biomarkers hold great promise due to their noninvasiveness, ability to record and monitor disease evolution in real time, and predict prognosis and therapy response.

The main liquid biopsy biomarkers of BCa include CTCs, ctDNA, and exosomes from the blood and urine. CTCs are tumor cells that originate from the tumor site and diffuse into the bloodstream, representing the main mechanism for metastasis [9,10]. The presence of CTCs has been identified as a predictive indicator of BCa prognosis and disease stratification [11,12,13,14,15]. In addition, the amount of CTCs in blood is inversely correlated with the cancer-specific survival rate of patients with metastatic BCa [16]. However, the relevance of CTCs to non-muscle invasive bladder cancer (NMIBC) is still debatable [17]. Moreover, because of its limited diagnostic sensitivity, CTC detection may not be accurate enough for use as an initial BCa screening test [11,14]. Improvements in the methods used to detect and obtain sufficient numbers of CTCs to depict the global view of tumor alterations requires improvement. Advances in clinical and laboratory techniques that enable the detection of CTCs at different time points may allow real-time surveillance of dynamic changes of BCa and may remarkably enhance our understanding of the metastatic cascade, thus facilitating the development of novel targeted therapy approaches. 

CtDNA is increasingly being investigated as a promising biomarker for various types of tumors [18]. Tumor cells release DNA molecules into the surrounding tissues, and the ctDNA is then transported into the bloodstream [19]. CtDNA is usually identified by somatic alterations that are specific for the tumor type [20] and is particularly suited for the detection of BCa, which exhibits a high mutational burden. The presence of ctDNA has been detected in the urine and plasma of BCa patients, and multiple studies have demonstrated the diagnostic and prognostic ability of ctDNA in both NMIBC and muscle invasive bladder cancer (MIBC) [21,22,23,24]. Furthermore, some studies have revealed drug-related ctDNA alterations in metastatic BCa patients, which emphasizes the importance of a ctDNA analysis for monitoring the therapy response [23,25,26]. Despite these findings, technological limitations have hindered progress in this area for decades. The relatively small amount of ctDNA fragments compared to the number of normal circulating DNA fragments means that accurate detection and quantification of ctDNA is difficult [27,28]. Of note, ctDNA can be released from dead tumor cells through apoptosis or necrosis; hence, genomic alterations derived from these cells may not completely reflect the biology of primary tumors or metastasis. 

To address the issues related to limited amounts of CTCs and ctDNA in liquid biopsy samples, exosomes have attracted interest. These extracellular vesicles (EVs), typically sized 40–160 nm, are secreted upon fusion of multivesicular bodies with the plasma membrane and are released into the surrounding body fluids. Exosomes contain a number of constituents, including nucleic acids (DNA, RNA, and miRNA); proteins; lipids; and metabolites (Figure 1), which strongly reflect the parental property, and are abundant in blood, making them a viable alternative to CTCs or ctDNA as biomarkers. Furthermore, due to their lipid bilayer, exosomes are extremely stable, are resistant to degradation by enzymes such as RNases (Figure 1), and can keep their contents intact for a longer time than CTCs. During tumorigenesis, single tumor cells continuously interact with each other and with normal host cells to boost tumor cell growth and survival, as well as tumor progression, angiogenesis, and metastasis [29]. Tumor cell-derived exosomes play a role in this tumor cell communication process through the transfer of their various ingredients from donor to recipient cells [30]. Consequently, genomic profiling of exosomes could be used to evaluate cancer diagnosis and prognosis. Indeed, a number of studies have shown that abnormal changes in exosomal molecules, including long noncoding RNAs (lncRNAs), proteins, and miRNAs, are associated with BCa tumorigenesis and may be useful diagnostic and prognostic biomarkers in liquid biopsies [31,32,33,34,35,36]. Although they are a promising source of cancer biomarkers, the use of exosomes is hampered by challenges associated with their isolation. Therefore, sensitive platforms that allow accurate isolation and detection are fundamental requisites for exosome research. The strengths and weaknesses of common liquid biopsy biomarkers, as well as their potential applications in BCa, are summarized in Table 1. 

### 1.3. Future Trends

Development of a noninvasive liquid biopsy biomarker with the potential to reduce the need for solid tumor biopsies will represent a momentous innovation in the field of precision medicine. In particular, liquid biopsy is a useful non-invasive tool for the discovery of urological cancer biomarkers [3], and urine-based research is particularly promising for BCa [47,48]. With respect to blood-based biomarkers, a major challenge is to evaluate the ability of peripheral markers to represent the complex tumor microenvironment. In this regard, urine exosomes appear to be more attractive, as they come from relatively close cancer tissues that share a common embryonal origin. Nonetheless, standardization of the isolation method among laboratories is a fundamental requirement to harness the potential usefulness of urinary exosomes in BCa. 

## 2. Exosomal miRNAs in Bladder Cancer Diagnosis and Prognosis

### 2.1. Exosomes and Exosomal miRNAs in Cancer 

Numerous studies have shown that exosomal contents, such as double-stranded DNA, various RNA species, and proteins, can be used as predictive biomarkers for cancer diagnosis and prognosis. Studies regarding the biomarkers have revealed that several RNAs possess advantages over DNA, due to their dynamic and fluctuating expression pattern (such as lncRNAs) corresponding with the internal needs of cells, thereby indicating that those RNAs have a high tissue- or disease-state-specific feature [49]. Furthermore, paradoxically, the DNAs existing in exosomes vary along with the different detection methods [50,51], which makes an elusive whether genomic DNA exists in exosomes. Among exosomal RNA species, miRNAs, a class of small, single-stranded, noncoding RNA molecules, play a role in virtually all biological pathways, including cell growth, proliferation, and differentiation, as well as immune responses, apoptosis, metabolism, and tumorigenesis. Exosomes provide a protective and highly stable source of miRNAs in body fluids, protecting them against degradation even under nonphysiological conditions. ExomiRs are reportedly resistant to multiple freeze-thaw cycles, are stable during long-term storage at room temperature, and remain stable for up to five years when stored at −20°C [29,52,53,54,55]. Their enhanced stability compared to proteins and other nucleic acids, both in the circulation and in fixed tissues, makes exomiRs well-suited to sampling and analysis [56,57]. Moreover, exosome secretion from malignant tissue is much higher than that from the corresponding normal tissue, and higher concentrations of exomiRs are typically detected in tumor liquid biopsy samples such as plasma, urine, and ascites [58,59,60,61]. Coupled with the stability of miRNAs, this increased load of circulating exosomes in the malignant state has enabled the identification of several potential exomiR biomarkers. Another attractive RNA molecule exists in exosome is lncRNA, which is larger in size and has less been studied than exomiRs—specifically, 534 versus 52 reports regarding miRNA and lncRNA, respectively, were searched in PubMed until January 2020 [62]. Nevertheless, granting exosomal lncRNA seems to be a promising biomarker for cancer; its application has some limitations, including an analysis only on the known lncRNAs, the false positives derived from nonspecific hybridizations, high variability for little expressed genes, and the sealed lncRNAs sequence variants. On the basis of these findings, the obvious superiority for the exomiRs as a noninvasive biomarker in liquid biopsies has been confirmed. 

MiRNAs are taken up by nearby or distal target recipient cells as a cargo of exosomes, reflecting a cell-to-cell communication method that can influence the pathogenesis of cancer [63,64]. Therefore, miRNAs carried by exosomes can provide information about dominant cells from which they are derived, as well as the target and cellular state, including potential therapy resistance. Recent evidence suggests that the transmission of onco-miRNAs via tumor cell-derived exosomes can promote tumor cell proliferation and invasion, as well as angiogenesis, distant metastasis, and remodeling of the tumor microenvironment [65].

MiRNAs are key regulators of gene expression in cancer, functioning as either tumor-suppressors or oncogenes depending on the target mRNA, and play a constructive role in tumorigenesis, metastasis, and resistance to diverse treatments [66,67]. Cancer cell-released exomiR-21, exomiR-23, exomiR-29, exomiR-103, and exomiR-210 promote tumor proliferation, angiogenesis, and migration [65,68,69,70,71,72,73,74]. In particular, exo-miR-21 may be a promising biomarker for many types of cancer. The roles of exomiRs in cancer are depicted in Figure 2.

### 2.2. Exosomal miRNAs as Potential Biomarkers for Bladder Cancer 

As exomiRs are one of the major components of exosomes and play a functional role in cell-to-cell communication, many studies of urinary exosomal biomarkers have focused on exomiRs. A study examining miRNAs in matched tumor tissue, plasma, urine exosomes, and white blood cells from patients with bladder cancer found that miR-4454, miR-205-5p, miR-200c-3p, miR-200b-3p, miR-21-5p, miR-29b-3p, and miR-720 /3007a were common to NMIBC tissues and urinary exosomes [35]. Although these results were from a small cohort (*n* = 16), they support the hypothesis that exomiRs reflect the molecular signals of parental tumor cells, thereby serving as desirable biomarkers for tumor characterization. The potential use of exosomes and exomiRs as diagnostic tools in BCa requires further investigation; however, increasing evidence suggests they have great potential. Previous clinical studies exploring the use of exomiRs as potential BCa biomarkers are summarized in Table 2.

#### 2.2.1. Diagnostic Markers

Several studies have reported exomiRs in the blood or urine as promising diagnostic biomarkers of BCa. One study using a device composed of nanowires to enable the collection of EVs with higher efficiency than that of the conventional ultracentrifugation method found that the levels of 22 and three exomiRs were upregulated and downregulated, respectively, in BCa urine [75]. Another microarray-based study found that the levels of 23 exomiRs were downregulated and those of three exomiRs were upregulated in the urine of high-grade BCa patients. An additional real-time PCR analysis then revealed that miR-375 and miR-146a could distinguish high- and low-grade BCa patients from healthy controls, respectively [76]. However, further studies are required to confirm the utility of these miRNAs in BCa diagnosis. In a more comprehensive study, five miRNAs (miR-155-5p, miR-15a-5p, miR-21-5p, miR-132-3p, and miR-31-5p) were expressed at significantly higher levels in urinary exosomes of BCa patients than those of the controls (all *p* < 0.0001) [77]. Among these urinary exomiRs, miR-21-5p showed the greatest potential for BCa detection, with an area under the receiver-operator characteristics curve (AUC) value of 0.900 (*p* < 0.0001) and a sensitivity and specificity of 75.0% and 95.8%, respectively. In the cohort examined, this level of diagnostic performance was better than that of urine cytology (sensitivity of 44.4% and specificity of 100%; Youden’s index: 0.708 versus 0.444), suggesting that miR-21-5p in urinary exosomes could be a single biomarker to detect the early stage and negative urine cytology of BCa [77]. 

Most biomarker studies have used a miRNA panel to improve the accuracy of BCa diagnosis. A quantitative reverse transcription PCR analysis revealed that the levels of four exomiRs (miR-21, miR-93, miR-200c, and miR-940) were higher in BCa urine samples than in healthy control samples, with a sensitivity and specificity of 88% and 78% (AUC = 0.888) [78]. Similarly, a regression model consisting of three urinary exomiRNAs (miR-30a-5p, let-7c-5p, and miR-486-5p) to discriminate between BCa and controls produced an AUC value of 0.70 [79]. In other BCa studies, analyses of a panel of six miRNAs in serum samples and a panel of seven miRNAs in urine samples produced AUC values of 0.899 and 0.923, respectively; the corresponding sensitivities of these panels for the detection of BCa stages Ta, T1, and T2–T4 were 90.0%, 84.9%, and 89.4%, respectively, and 86.4%, 93.0%, and 97.8%, respectively. All of these values were significantly higher than the sensitivities of urine cytology, which were 13.3%, 30.3%, and 44.7% and 53.1%, 62.8%, and 72.4%, respectively (all *p* < 0.001) [80,81]. Urquidi et al. developed a urine-based panel of 25 miRNAs to detect the presence of BCa, which exhibited a sensitivity of 87.0% and a specificity of 100.0% (AUC = 0.982), indicating that the panel could facilitate the noninvasive evaluation of patients under investigation for BCa [82]. 

Although the miRNA panels described above showed acceptable sensitivities for BCa detection, the complexities associated with the evaluation of multiple markers mean that they are not suitable for clinical application. A clinically useful biomarker should be not only accurate but, also, simple and cost-effective. A diagnostic model using a combined index of the levels of two urinary miRNAs (miR-99a and miR-125b), both of which were reportedly downregulated in BCa urine, showed a sensitivity of 86.7%, a specificity of 81.1%, and a positive predictive value of 91.8% [83]. Moreover, a model using the level of miR-125b alone to discriminate between high- and low-grade BCa produced similar results (sensitivity, 81.4%; specificity, 87.0%; and positive predictive value, 93.4%) [83]. In our previous study using 543 urine samples, we found that the urinary miR-6124 to miR-4511 ratio was considerably higher in BCa than in hematuria or pyuria and enabled the discrimination of BCa from hematuria with a sensitivity of >90.0% (*p* < 0.001), suggesting that it may be a useful noninvasive diagnostic tool to reduce unnecessary cystoscopies in patients with hematuria undergoing evaluation for BCa [36]. 

Urinary biomarkers could also be used to correctly identify MIBC patients that require radical cystectomy, a treatment that severely affects the quality of life. Although no such biomarkers have been identified to date, Baumgart et al. identified three miRNAs (miR-30a-3p, miR-99a-5p, and miR-137-3p) that were upregulated and two (miR-141-3p and miR-205-5p) that were downregulated in exosomes secreted by invasive BCa cells versus those secreted by noninvasive BCa cells [84]. However, the suitability of these miRNAs as liquid biomarkers to discriminate between NMIBC and MIBC requires further analysis.

#### 2.2.2. Prognostic Markers

The urinary level of miR-214 is an independent predictor of NMIBC recurrence (hazard ratio, 2.011; 95% confidence interval, 1.027 to 3.937; *p* = 0.041), indicating the prognostic ability of this miRNA [85]. However, other studies have reported that diverse miRNAs, such as serum miR-152 and urinary miR-22-3p and miR-200a-3p, could also provide information on the recurrence risk of NMIBC [80,81].

Similar to their use to identify diagnostic markers of BCa, miRNA panel studies have also been performed to identify prognostic markers. Urinary profiling using a panel of six miRNAs to distinguish patients with NMIBC recurrence produced an AUC value of 0.85 [86]. The performance of this panel was validated in an independent cohort and detected recurrence with a high sensitivity (88.0%) and sufficient specificity (48.0%) (AUC = 0.740). The panel was particularly suitable for the detection of clinically significant diseases, such as T1 stage (AUC = 0.920) and high-volume disease (AUC = 0.810), suggesting that it may be useful for reducing the rate of invasive and costly cystoscopies during BCa surveillance [86]. In another study, a panel of four miRNAs (miR-422a-3p, miR-486-3p, miR-103a-3p, and miR-27a-3p) was able to predict the development of MIBC with AUC values of 0.894 and 0.880 for the training and validation sets, respectively, both of which were significantly higher than those of the BCa grade and urine cytology (both *p* < 0.05) [87]. Moreover, a Kaplan-Meier analysis showed that lower expression levels of miR-486-3p and miR-103a-3p were associated with lower overall survival rates of MIBC patients (both *p* < 0.05) [87].

#### 2.2.3. Treatment Prediction Markers

ExomiRs released from cancer cells or noncancerous cells, which have different effects on target cells in the tumor microenvironment, can affect the development of drug resistance [88]. ExomiRs can induce resistance to cytotoxic and molecular target-specific drugs [89]. MiR-155 can reportedly promote resistance to doxorubicin and paclitaxel in multiple cancer types [88,90,91,92]. In addition, some miRNAs seem to regulate therapy resistance in a tumor-specific manner; for example, miR-128-3p and miR-425-3p act as drug resistance mediators in colorectal and lung cancer, respectively [93,94]. 

BCa responses to chemotherapy are also associated with exomiRs. Fanous et al. examined the roles of exomiRs in BCa chemoresistance to cisplatin, gemcitabine, and cisplatin/gemcitabine and found that exosomes from chemo-resistant CUB III cells exhibit distinct miRNA profiles [95]. Among the 759 miRNAs profiled, 16 were differentially expressed (10 downregulated and 6 upregulated) in almost all CUB III-resistant sublines relative to their parental line. Of these, miR-Let-7i-3p was the most significantly downregulated exomiR, while miR-21-5p was the highest upregulated exomiR, suggesting that these exomiRs could be used as predictive biomarkers of treatment responses and/or therapeutic targets to augment drug responses [95]. Although studies of exomiRs as treatment prediction markers are limited, they have the ability to provide information about their location of origin, including the specific target, cellular state, and potential therapy resistance; therefore, it may be possible to monitor and regulate tumor resistance, as well as achieve personalized therapy, using exomiR biomarkers.

## 3. Challenges Associated with the Use of Exosomal miRNAs as Biomarkers

Despite the great benefits of exomiRs in cancer management, the field faces many scientific and technical hurdles. First, there is no consensus method for isolating and purifying exomiRs, and the effect of using disparate methods on miRNA expression profiling is unclear. A standard methodology is needed to ensure single EV isolation from complex body fluid. However, well-characterized, publicly available reference standards for optimizing the collection, storage, and processing of EV-containing body fluids are still limited [96,97,98]. Second, it is not easy to separate a specific tumor-derived exosome, as body fluids are an opulent source of exosomes that originate from both tumor and normal cells. In addition, some protein fragments are enclosed in urine and serum during the isolation process, which may interfere with the results [99]. Third, a procedure for normalizing exomiR expression levels using an appropriate endogenous housekeeping miRNA has not yet been established. Although a number of studies have investigated the use of various miRNA controls for normalization, none have been identified as acceptable. As an alternative to the use of an internal control, we have used the ratio of up- and downregulated miRNAs to examine the use of urinary cell-free miRNAs as diagnostic biomarkers of BCa [36]. Finally, sample-related biases in exomiR expression remain a challenge. Analyses of small sample sizes can produce unreliable results, and accurate data comparisons require large-scale studies of prospective cohorts. Overall, to obtain clinically applicable results, a consensus has to be reached regarding optimal exomiR isolation methods, and there is a need to develop computational tools that permit the accurate analyses of single EV-associated exomiRs to identify their cells of origin and limit data heterogeneity.

## 4. Concluding Remarks

Analyses of exosomal cargoes in liquid biopsy samples revealed the diagnostic and prognostic potentials of exomiRs as noninvasive tools to identify the genetic landscape of tumors. Numerous studies have explored the profiles of exosomes and exomiRs in BCa and attempted to characterize their potential clinical applications. However, exosomal research in cancer, especially BCa, is still in the early stages. Like other new biomarkers, cancer-related exosome research faces many challenges. Nonetheless, exomiRs are highly promising biomarker candidates that may aid BCa diagnosis and prognosis.

## Figures and Tables

**Figure 1 ijms-22-01713-f001:**
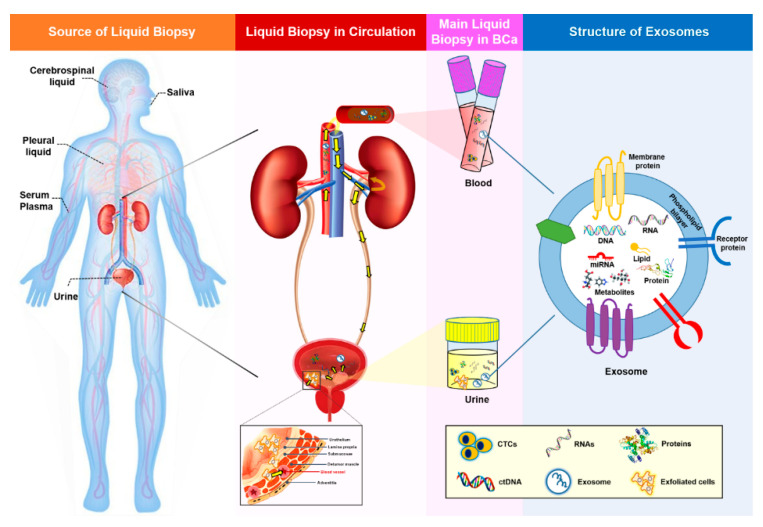
Summary of liquid biopsy and liquid biopsy biomarkers. Liquid biopsy samples represent all body fluids, including urine, serum, plasma, saliva, cerebrospinal fluid, and pleural fluid. In BCa, urine (which intimately contacts the tumor), serum, and plasma are used widely for the detection and surveillance of the disease. Genetic information from tumor cells is spread through the circulation (yellow arrows in the second image indicate the circulation in the urinary system), thereby favoring the detection of liquid biopsy biomarkers in the blood and urine. Common biomarkers present in liquid biopsies include CTCs, ctDNA, and exosomes. Moreover, exfoliated cells derived from a tumor can also be found in urine. Tumor cells continuously influence surrounding or distal sites by transmission of cancerous signals. Exosomes act as important mediators of cell-to-cell communication by transferring their contents, including DNA, RNA, miRNA, lipids, proteins, and metabolites. Due to their lipid bilayer structure, exosomes are extremely stable and can resist degradation by enzymes such as RNases. BCa, bladder cancer; CTC, circulating tumor cell; and ctDNA, circulating cell-free tumor DNA.

**Figure 2 ijms-22-01713-f002:**
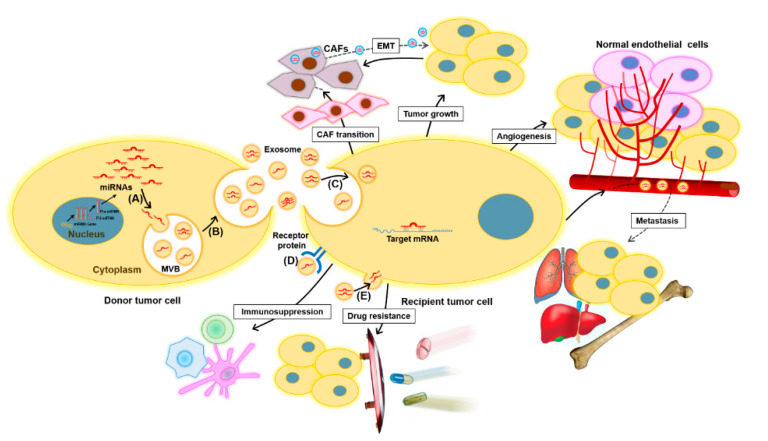
Functions of exomiRs in cancer. ExomiRs play multiple roles in tumor initiation and development. Tumor cells may transfer exomiRs to the surrounding normal or tumor cells as follows: (A) the sorting of miRNAs into exosomes during MVB formation, (B) the release of exosomes into the extracellular space, (C) the delivery of exomiRs to recipient cells by endocytosis, (D) the binding of exosomes to receptors to activate specific signaling pathways, and (E) the fusion of exosomes with the plasma membrane of recipient cells. Subsequently, the transmitted exomiRs can regulate tumor cell growth, angiogenesis, and metastasis by binding to their downstream targets. Drug-resistant tumor cells can transmit their resistant phenotype to drug-sensitive tumor cells through the transfer of exomiRs that confer chemoresistance. Moreover, tumor-derived exomiRs play a key role in exchanging information between tumor cells and immune cells, such as macrophages, T cells, and dendritic cells, to suppress the immune response and promote tumor progression. Tumor-derived exomiRs are also instrumental modulators of the tumor microenvironment, inducing CAF transition, thus facilitating tumorigenesis and tumor development. On the other hand, fibroblast-derived exomiRs are also transferred to tumor cells and can contribute to EMT. CAF, cancer-associated fibroblast; EMT, epithelial–mesenchymal transition; ExomiR, exosomal miRNA; and MVB, multivesicular body.

**Table 1 ijms-22-01713-t001:** Summary of advantages and disadvantages of liquid biopsy biomarkers.

Type of Liquid Biopsy Sample	Origin	Detection Methods	Advantages	Disadvantages	Applications in BCa	References
CTC	Primary tumors; Metastatic sites.	Microfilters; Immunocytochemistry; Immunomagnetic assays; CellSearch System; CytoTrack; Epic CTC Platform; HD-CTC; RT-PCR.	High specificity; Non-invasive analysis; Represent genomic information from the tumor or metastatic sites in real time; Can observe response to therapeutic treatments.	Low sensitivity to monitor tumors, particularly early stage tumors.	Valuable prognosis marker associated with disease-free survival in metastatic BCa; Useful for deciding therapeutic approaches, since it is strongly correlated to tumor stage, histological grade, and metastasis.	[9,11,12,13,14,15,16,37,38]
ctDNA	Released from viable tumors by activating secretion, apoptosis, or necrosis; Destruction of CTCs.	PCR; Pyrosequencing; RT-PCR; NGS; ddPCR; BEAMing technology; CAPP-Seq.	High sensitivity; Non-invasive analysis; Stability in variable degradation microenvironments; Can be used to detect mutations related to carcinogenesis.	Require a specific method to distinguish tumor origin DNA from healthy origin DNA, since both release ctDNA; Low yield in plasma; Low sensitivity to monitor tumors, particularly early stage tumors.	Genomic alterations such as specific mutations, deletions, and methylation variations detected in ctDNA from BCa urine and plasma are correlated to disease recurrence and progression; Alterations in ctDNAs with differential sensitivity to therapeutic agents could be used as markers of therapy response in metastatic patients.	[12,13,15,17,21,22,23,24,25,26,39,40,41,42,43]
Exosome	Primary tumors; Cells in various body fluids	Real-time PCR; ddPCR; NGS; Microarray; Western blotting; ELISA.	Non-invasive analysis; High stability in variable degradation microenvironments; Better sensitivity and specificity than CTC and ctDNA in various bio-specimens.	Challenging protocols to analyze genomic materials in exosomes	Urinary exosomal miRNAs show potential for BCa detection; Long non-coding RNAs and proteins in urinary exosomes are proposed to be enriched in BCa.	[28,31,32,33,34,35,36,44,45,46]

BCa, bladder cancer; CAPP-Seq, cancer personalized profiling by deep sequencing; CTC, circulating tumor cell; ctDNA, circulating cell-free tumor DNA; ddPCR, digital droplet PCR; HD-CTC, high-definition CTC; NGS, next-generation sequencing; and RT-PCR, reverse-transcription PCR.

**Table 2 ijms-22-01713-t002:** List of exomiRs identified in BCa.

Markers	Biological Source	Regulation	Clinical Significance	Reference
miR-1285-3p, miR-142-3p, miR-16-1-3p, miR-195-3p, miR-196b-5p, miR-23b-3p, miR-28-5p, miR-299-3p, miR-3155a, miR-3162-5p, miR-3678-3p, miR-4283, miR-4295, miR-4311, miR-4531, miR-492, miR-5096, miR-513b-5p, miR-5187-5p, miR-601, miR-619-5p, miR-92a-2-5p	Urine	Up	Diagnosis	[75]
miR-375, miR-146a	Urine	Up	Diagnosis	[76]
miR-155-5p, miR-15a-5p, miR-21-5p, miR-132-3p, miR-31-5p (especially miR-21-5p)	Urine	Up	Diagnosis	[77]
Four-miRNA panel: mir-21, miR-93, miR-200c, and miR-940	Urine	Up	Diagnosis	[78]
Three-miRNA panel: miR-30a-5p, let-7c-5p and miR-486-5p	Urine	Up	Diagnosis	[79]
Six-miRNA panel: miR-152, miR-148b-3p, miR-3187-3p, miR-15b-5p, miR-27a-3p and miR-30a-5p	Serum	Up	Diagnosis	[80]
Seven-miRNA panel: miR-7-5p, miR-22-3p, miR-29a-3p, miR-126-5p, miR-200a-3p, miR-375 and miR-423-5p	Urine	Up	Diagnosis	[81]
Twenty-five-miRNA panel: miR-652, miR-199a-3p, miR-140-5p, miR-93, miR-142-5p, miR-1305, miR-30a, miR-224, miR-96, miR-766, miR-223, miR-99b, miR-140-3p, let-7b, miR-141, miR-191, miR-146b-5p, miR-491-5p, miR-339-3p, miR-200c, miR-106b, miR-143, miR-429, miR-222 and miR-200a	Urine	Up	Diagnosis	[82]
Ratio of miR-6124/miR-4511	Urine	Up	Diagnosis	[36]
miR-30a-3p, miR-99a-5p, miR-137-3p	Cells	Up	Discrimination of MIBC from NMIBC	[84]
miR-141-3p, miR-205-5p	Cells	Down	Discrimination of MIBC from NMIBC	[84]
miR-99a, miR-125b	Urine	Down	Diagnosis	[83]
miR-152	Serum	Up	Prediction of NMIBC recurrence	[80]
miR-22-3p, miR-200a-3p	Urine	Up	Prediction of NMIBC recurrence	[81]
Six-miRNA panel: miR16, miR200c, miR205, miR21, miR221 and miR34a	Urine	Up	Prediction of NMIBC recurrence	[86]
Four-miRNA panel: miR-422a-3p, miR-486-3p, miR-103a-3p and miR-27a-3p	Serum	Up	Prediction of MIBC survival	[87]
miR-214	Urine	Down	Prediction of NMIBC recurrence	[85]
miR-21-5p	Cells	Up	Prediction of response to chemotherapy	[95]
miR-Let-7i-3p	Cells	Down	Prediction of response to chemotherapy	[95]

BCa, bladder cancer; MIBC, muscle invasive bladder cancer; NMIBC, non-muscle invasive bladder cancer; and exomiRs: exosomal miRNAs.

## Data Availability

The data presented in this study are available on request from the corresponding author.

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
