# Peer review of "Role of Exosomal miRNA in Bladder Cancer: A Promising Liquid Biopsy Biomarker"

_ijms, 2021, doi:10.3390/ijms22041713_

Round 1

Reviewer 1 Report

It is a comprehensive review of the current knowledge on liquid biopsy and exosomal miRNA in bladder cancer. The authors did a good job on collecting and organizing available information for using exosomal miRNAs as bladder cancer biomarkers. Following are my comments:

1. At the beginning of section 2. Exosomal miRNAs in Bladder Cancer Diagnosis and Prognosis, the authors should use a paragraph to further elaborate the properties of exosomes, and why miRNAs in particular, rather than DNA/LncRNA etc, are of great value in BCa diagonosis and progosis.

2. Figure 2 is cropped out of the page, could not see the whole figure.

3. Please rearrange the format of references into a uniformed style.

Author Response

Response to Reviewer 1 Comments

Point 1: At the beginning of section 2. Exosomal miRNAs in Bladder Cancer Diagnosis and Prognosis, the authors should use a paragraph to further elaborate the properties of exosomes, and why miRNAs in particular, rather than DNA/LncRNA etc, are of great value in BCa diagonosis and progosis.

Response 1: We have added some evidences that why are exomiRs rather than DNA or lncRNAs according to reviewer’s comment. These have been cited in page 5 and page 6, in red colour. Additionally, the yellow highlighted point (page 6) is that we have mentioned the superiority of exomiRs mentioned in our previous manuscript.

Point 2: Figure 2 is cropped out of the page, could not see the whole figure.

Response 2: The figure has been rearranged now.

Point 3: Please rearrange the format of references into a uniformed style.

Response 3: We have rearranged the format according to reviewer’s comment.

Reviewer 2 Report

This study was well-written. The present manuscript could be accepted. 

Author Response

Thank you for the reviewer's work. 

Reviewer 3 Report

This paper provide us a new review about the role of exosomes  in bladder cancer. Nevertheless, it focus on main cargo of exosomes namely miRNAs. The authors explain the function of miRNAs as markers of this tumor and very nicely explain thy function in the pathogenesis of this cancer and their ability to transfer signals from tumor cells to suborned cell. Finally, they claim that by using   liquid biopsy  it became an easy non invasive and practical diagnostic approach. The paper is nicely written and bring us a nice new approach for detecting biomarkers for bladder carcinoma

Author Response

Thank you for the reviewer's work.